# Modelling the effects of the repellent scent marks of pollinators on their foraging efficiency and the plant-pollinator community

**Elise Verrier** *, **Emmanuelle Baudry, Carmen Bessa-Gomes**

Ecologie Systématique Evolution, UMR 8079, CNRS, AgroParisTech, Université Paris Saclay, Orsay, France

* elise.verrier@laposte.net

## Abstract

Pollinator insects forage in complex and unpredictable resource landscapes, often using social information from congeneric individuals to acquire knowledge about their environment. It has long been recognized that this process allows them to exploit floral resources more efficiently and thus increase individual fitness. However, by creating correlations between the behaviors of pollinators within a population, this could also indirectly influence the entire plant-pollinator community. One type of social information used by pollinators is the scent mark left on the corolla of flowers by previous visitors, which can be used as a cue to avoid recently depleted resources. We developed a spatially explicit agent-based model to examine the effects, at both individual and community levels, of pollinators using these scent marks. The model simulates a population of pollinators foraging on flowers in a continuous 2D space in which we can vary the density of pollinators. We showed that the use of scent marks as a source of information significantly increased the foraging efficiency of pollinators except when competition between pollinators was very low. At the community level, this also resulted in a marked homogenization between floral resources within the landscape: in the absence of scent marks, the coefficient of variation of the remaining nectar quantity per flower strongly increased with greater pollinator competition, but it remained low at all levels of competition when scent marks were used by the pollinators. Finally, the use of scent marks markedly decreased the number of pollinator flower visits, especially at high levels of pollinator competition, which can potentially reduce the pollination service.

## 1. Introduction

In many species, individuals can increase their knowledge about their environment using the social information provided by congeneric individuals [1, 2]. The social information concept was initially developed for vertebrates, and it has been evoked in a diversity of context, from selection of breeding sites in kittiwake to home range formation in wolves and coyotes [3, 4]. The concept has since been extended to insects, particularly pollinators foraging in complex and unpredictable environments [5, 6].

**Data Availability Statement:** All the components files of the agent based model will be available in a public GitLab repository.

**Funding:** E.V. is funded by the French Ministry of Research and the SEVE Doctoral School. This work is part of the "Investments d'Avenir" Programme overseen by the French National Research Agency (ANR) (LabEx BASC; ANR-11-LABX-0034). The funders had no role in study design, data collection and analysis, decision to publish, or preparation of the manuscript.

**Competing interests:** The authors have declared that no competing interests exist.

For an insect pollinator, flowering plants are a rich source of easily accessible reward in the form of nectar and pollen. Nevertheless, foraging is an extremely complex task, since the small amount of resource per flower makes it necessary to visit a very large number. For this process to be efficient, the resource gain must exceed the energy cost of moving between flowers. To exploit floral resources optimally, pollinators must therefore consider a large number of factors such as the abundance and location of the flowers, their ease of handling, and the amount of reward produced by the flowers, which often changes rapidly due to the consumption by other pollinators.

To obtain information about these different factors, pollinating insects can interact directly with their environment to acquire personal information (for example the localization of flowers) but they can also interact with or observe other individuals and acquire social information [7, 8]. Social information is often divided between, on the one hand, signals shaped by natural selection to transmit information and, on the other, inadvertently provided cues consisting of elements that are inevitably produced by organisms [8]. For example, the well-known waggle dance of honeybees is a signal that allows other members of the hive to acquire information about the distance and direction of flower patches with available nectar and pollen [9, 10]. By contrast, bumblebees have been shown to be more effective at selecting flower types with high rewards using the inadvertent clues provided by conspecifics on these flowers [11, 12].

Several species of pollinators use repellent scent marks left by previous visitors on the corolla of flowers. This phenomenon has been well studied in social pollinators like honeybees and bumblebees but has also been described in solitary bees and is likely to be widespread among the pollinator community [13–18]. These scent marks consist of short-lived odors found on the cuticula of pollinators. Over time, the scent mark evaporates and fades while the nectar is replenished until the flowers are no longer avoided by pollinators. The detection of repellent scent marks is supposed to improve the foraging efficiency of pollinators by allowing them to identify and avoid depleted flowers. This ability to avoid recently emptied flowers is supposed to be particularly useful when both flower handling time and pollinator competition are high [19–21]. Indeed, if most flowers are full of nectar, then it is less necessary to avoid the few empty flowers, but conversely, if the pollinator density is high and the proportion of empty flowers is important, the use of scent marks helps pollinators to increase their foraging profit.

In general, the use of social information leads to correlations between the behaviors of individuals who produce the information and those who use it [22]. In the case of repellent scent marks, there is a negative correlation, since the probability that a flower will be foraged at a given time by a given insect decreases if another insect foraged it shortly before. These correlations are expected to modify the distribution of nectar resources in the landscape. We propose that the negative correlation induced by repellent scent marks will homogenize the amount of nectar available from the different flowers in the landscape. Moreover, the use of scent marks will tend to reduce the number of visits to flowers by pollinators, thus potentially diminishing the efficiency of the pollination service.

We developed an agent-based model to examine at the individual level how the use of scent marks affects pollinator foraging efficiency as a function of the level of competition in the population, and at the community level, how it impacts the distribution of resources in the landscape and the efficiency of pollination services for flowers.

## 2. Materials and methods

We developed an agent-based model to examine how the use of repellent scent marks as public information affects both pollinator foraging efficiency and the flower community. Our model

was designed to simulate pollinators foraging in a meadow with a specific density of flowers. The pollinators exploit a stable but unknown environment and compete with conspecifics. At the end of the simulation, we record the mean nectar quantity collected by pollinators during the simulation, the mean nectar quantity remaining in the flowers, the mean number of visits per flower during the simulation, and the pollen transfer (estimated as the number of flowers receiving the pollen of a focal flower), as well as the coefficient of variation of these four variables. The model is coded in Python 3.8 [23] and is available on GitLab (https://gitlab.com/EliseVerrier1/abm-pollinators-scent-mark). A detailed version of the model description based on the ODD (Overview, Design concepts, Details) protocol [24, 25] is available in S1 File along with a detailed description of the model parameters, variables, and their values (Tables S1.1-S1.4 in S1 File).

## 2.1. Model entities and variables

The model is composed of three entities: the meadow (environment occupied by the agents) and two classes of agents, namely flowers and pollinators. The meadow is a continuous torus in 2D. The environment is characterized by its *topology*, i.e., a square with sides set by the user (*size_map*). In our model, the environment keeps track of the discrete time during the simulation with the dynamic variable *step*. The time that the pollinator spends in the flower while visiting it (*cost*) is also considered to be an environmental variable.

Flowers are immobile agents characterized by the amount of their nectar and the concentration of the scent mark left on their corolla at each step of the simulation. Both variables are a function of the time since the last pollinator visit, the speed of nectar refill, and the speed of the disappearance of the scent mark left by the pollinator. In our present analysis, we chose to consider the scent mark as a reliable signal of nectar refill. Flowers will also record the number of insects visiting them.

Pollinators are mobile agents characterized by their use or non-use of scent marks as repellent cues. Their foraging performance is examined in light of two variables: the amount of nectar collected, and the number of flowers visited. In our simulations, all pollinators within any individual simulation are identical regarding these traits.

## 2.2. Process overview and scheduling

In this discrete time model, one step represents the time required for one action by the pollinator. The foraging duration is 1,000 steps. At the start of the simulation, the flowers are full of nectar, and the pollinator nectar harvest is set to 0. After initializing the environment, the flowers are randomly positioned. The effect of the number of flowers was examined in the preliminary sensitivity analysis and was fixed at 250 for the simulation experiments. To model various levels of competition, our simulation experiments examined a range of values for the number of pollinators per flower. The pollinators are randomly placed on the map. Due to the random distribution and stochasticity of the model, the possibility that two agents of the same category (i.e., two flowers or pollinators) are located at the same place is insignificant. Nevertheless, we decided that multiple pollinators could not be simultaneously present on the same flowers.

**2.2.1. Pollinator process scheduling.** The process scheduling for the pollinator is organized in five major classes of actions (summarized in Fig 1). When the pollinator is on a flower, its behavior depends on whether it uses the scent mark. If the pollinator uses this source of public information, the probability of visiting the flower is the outcome of a Bernoulli trial with the probability set to correspond to the concentration of scent marks on the flower. In the case of success, the pollinator will visit the flower and take the nectar; otherwise, it will move at the next time step. If the pollinator does not use social information, it will always visit the

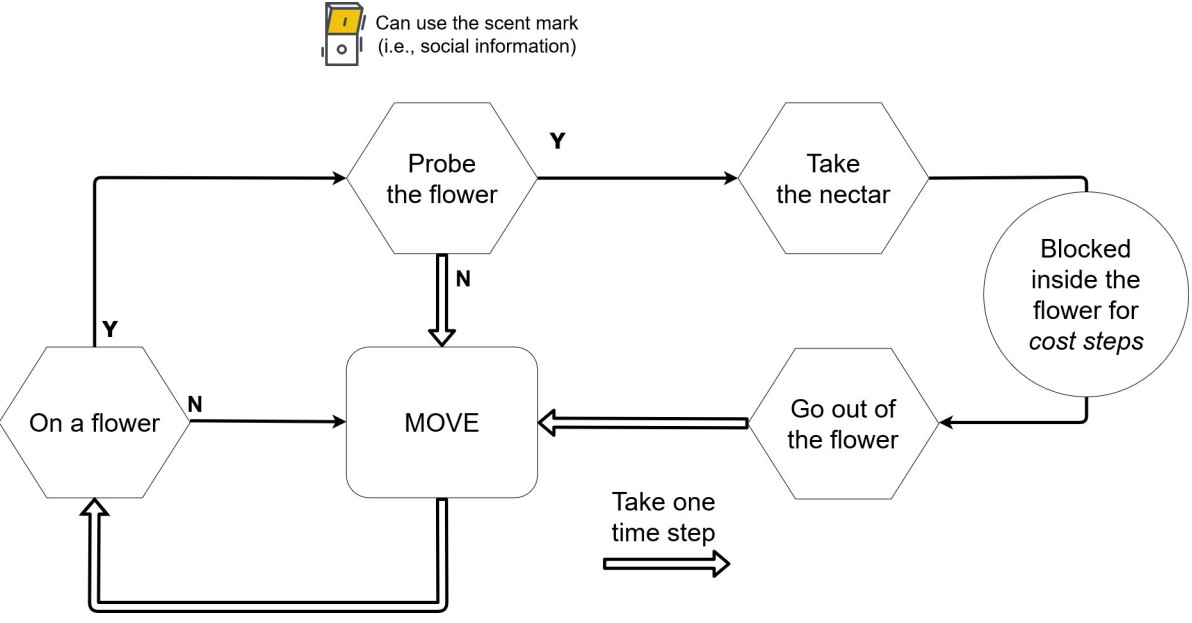

**Fig 1. Scheduling of the pollinator.**

flower. When a pollinator enters the flower, it will harvest all the available nectar, and this quantity will be added to the total amount of collected nectar.

In our model, the major cost associated with exploiting a flower is the time involved, a cost likely to depend on flower complexity. The pollinator will stay in the flower for a fixed period.

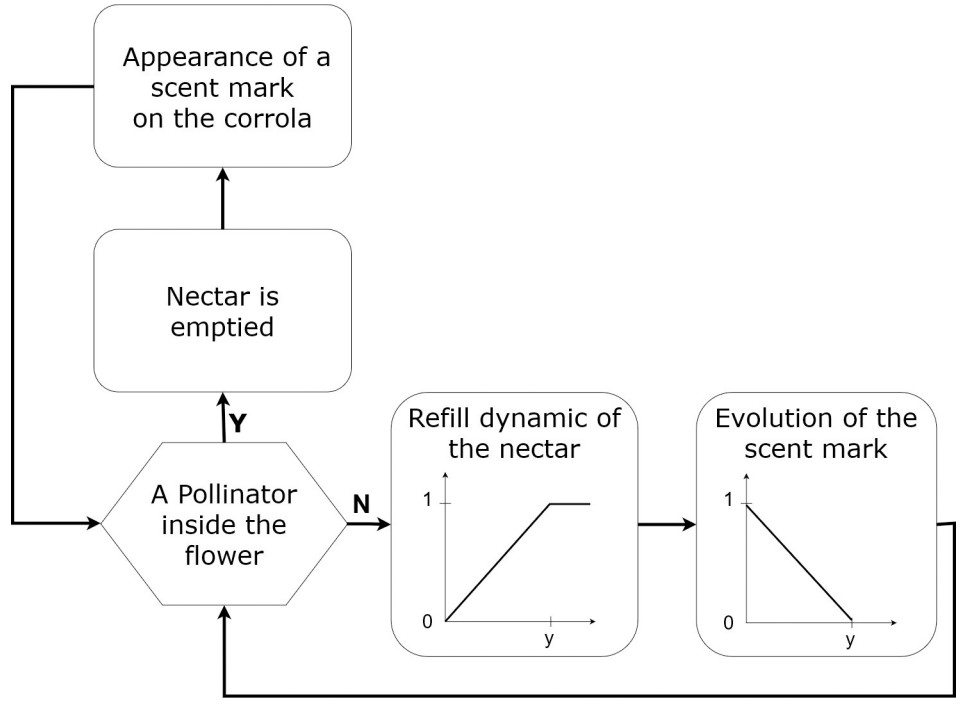

**Fig 2. Scheduling of the flower.**

In our study, all flowers are homogenous, and the time steps spent on flowers are the same for all individuals on a flower.

When the pollinator is not on a flower, it will move. In the case of a novel free flower within the pollinator's detection range (a fixed detection radius) with no other pollinator above or inside it, the pollinator will move there. If there is more than one such flower, one is selected randomly. If there is no free flower, it will move in a direction that diverges from the previous direction by an angle taken from a Gaussian with expected value zero and variance depending on the degree of inertia.

**2.2.2. Flower process scheduling.**   When all pollinators have completed their action, the scent mark and nectar quantity of every flower are updated. Both events are linearly dependent on the time since last visit (Fig 2) [20]. Following the pollinator's presence within the flower, the nectar quantity is set to 0, whereas the scent mark is set to 1. As the time since last visit increases, the amount of nectar accumulates, and the scent mark fades.

## 2.3. Sensitivity analyses

The impact of each parameter on the quantity of collected nectar was first assessed using sensitivity analysis by simulation that explored a range of parameters (Table 1). We ran 10,000 simulations, and for each simulation, parameters were assigned a value that was randomly taken from the variation interval. Their impact was studied by performing a random forest analysis in R version 4.0.3 [26] using the *randomForest* package [27]. We built 2,000 regression trees and examined three variables at each split.

The sensitivity analysis explained more than 95% of the variance. It is presented in detail in S2 File.

## 2.4. Simulation experiments

Sensitivity analysis informed our choice of model parameters. The amount of nectar collected increases linearly with the number of steps. As this parameter markedly influences the duration of the simulation, it was set to 1,000 steps. Because the coefficient of nectar refill has the greatest impact on the quantity of nectar collected (Fig S2.1 in S2 File), we chose to examine three contrasting values (0.001, 0.0025, 0.005), corresponding to 1, 2.5, and 5 refills per simulation, respectively. The time spent probing each individual flower (cost parameter in our model) will influence the amount of nectar collected and the advantage of using repellent scent marks. We decided to set the cost to a low value (5-time steps, i.e., the pollinator remains in

**Table 1. List of the model parameters and their variation range.**

| Type | Parameters | Interval | Value |
|---|---|---|---|
| Environment | Map size | [30–100] | 50 |
| | Number of steps in the simulation | [700–1300] | 1000 |
| Flowers | Number of flowers (as a function of meadow area) | [0.02–2] | 0.1 (250 flowers) |
| | Coefficient of nectar refill | [0.001–0.005] | [0.001, 0.0025, 0.005] |
| Pollinators | Pollinators per flower | 0.1–2 | [0.1, 0.3, 0.5, 1, 2] |
| Foraging behavior | Use of repellent scent marks | True or False | True or False |
| | Time spent probing each flower | [1–10] | 5 |
| Flower detection and movement | Detection radius | 1 | 1 |
| | Degree of inertia in pollinators' random movement | [0.01–0.4] | 0.1 |

The values exploited during sensitivity analysis appear in the "interval", while the column "value" presents values exploited in simulation experiments and default values for the remaining parameters.

each flower for 0.5% of the simulation) to focus on the consequences of competition on information use.

The intensity of competition experienced by pollinators depends on the ratio of pollinators to flowers, while the time spent locating the flowers depends on the relation between the number of flowers and the size of the simulated meadow. These parameters increase the mean standard error of the model by almost 50% (Fig S2.1 in S2 File). We chose to fix the number of flowers in relation to the meadow size and examine the impact of varying the number of competing pollinators. We set the maximal number of pollinators to 500, with the density of flowers and the size of the map being chosen in consequence.

The analyses were performed with R 4.0.3 [26] using the *Rmisc* [28] and *plyr* [29] packages. The graphs were created with the ggplot2 package [30].

## 3. Results

### 3.1 Efficiency of nectar collection by pollinators

**3.1.1 Impact of the cost.**   We first study the influence of the amount of time necessary to exploit a flower on pollinator efficiency. This time represents an exploitation cost that affects the amount of nectar collected and can be linked to the flower complexity. Our results showed that when the cost increases, it gives an advantage to individuals using the scent mark, even in a low competition context (Fig 3).

To keep the focus on the consequences of the competition, we decided to set the cost to the relatively low value of 5 (0.5% of the simulation time) in the rest of this study.

**3.1.2 Impact of the competition.**   Then, we analyzed the effect of scent mark use on the efficiency of pollinators' nectar collection for three flower nectar secretion rates and for relative

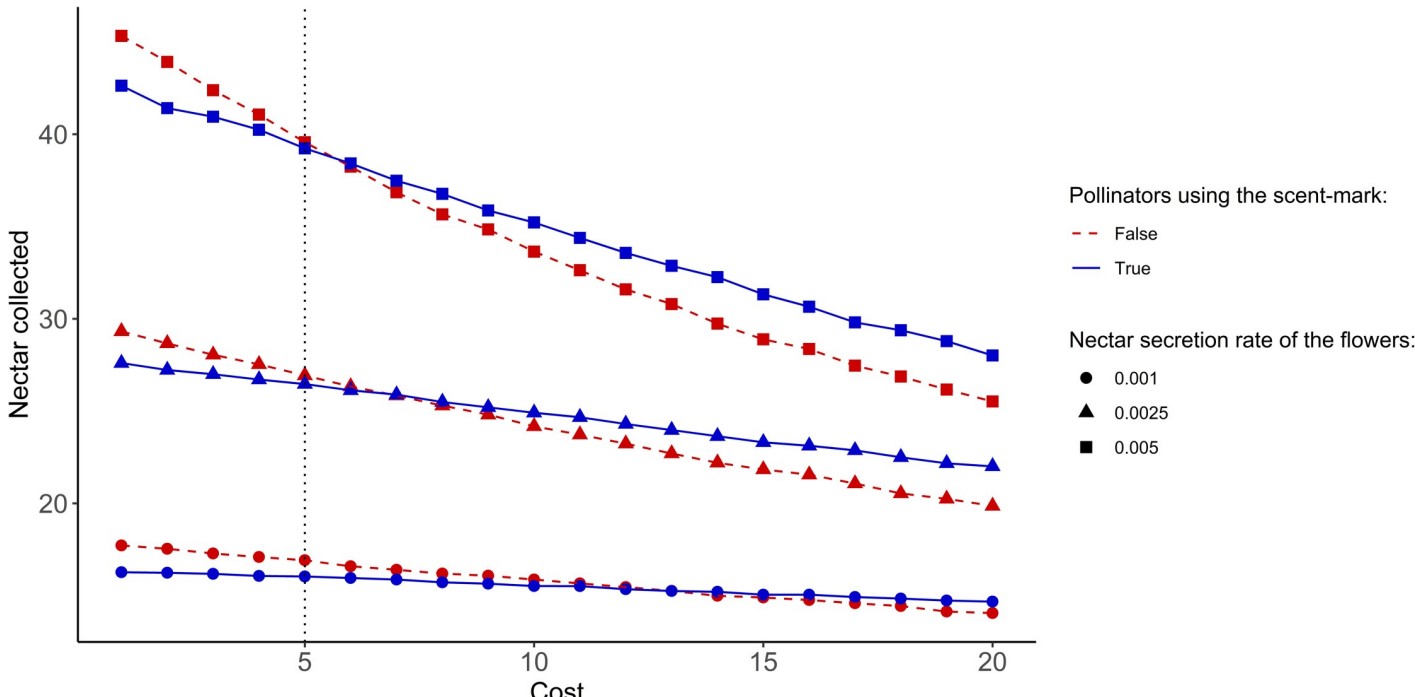

**Fig 3. Mean amount of nectar collected by pollinators during the simulation as a function of the cost.** The relative number of pollinators per flower was set to 0.1 (low pollinator density). The vertical dotted line represents the default value chosen for our analyses.

abundances of pollinators compared to flowers varying from 0.1 to 2 pollinators per flower (Fig 4).

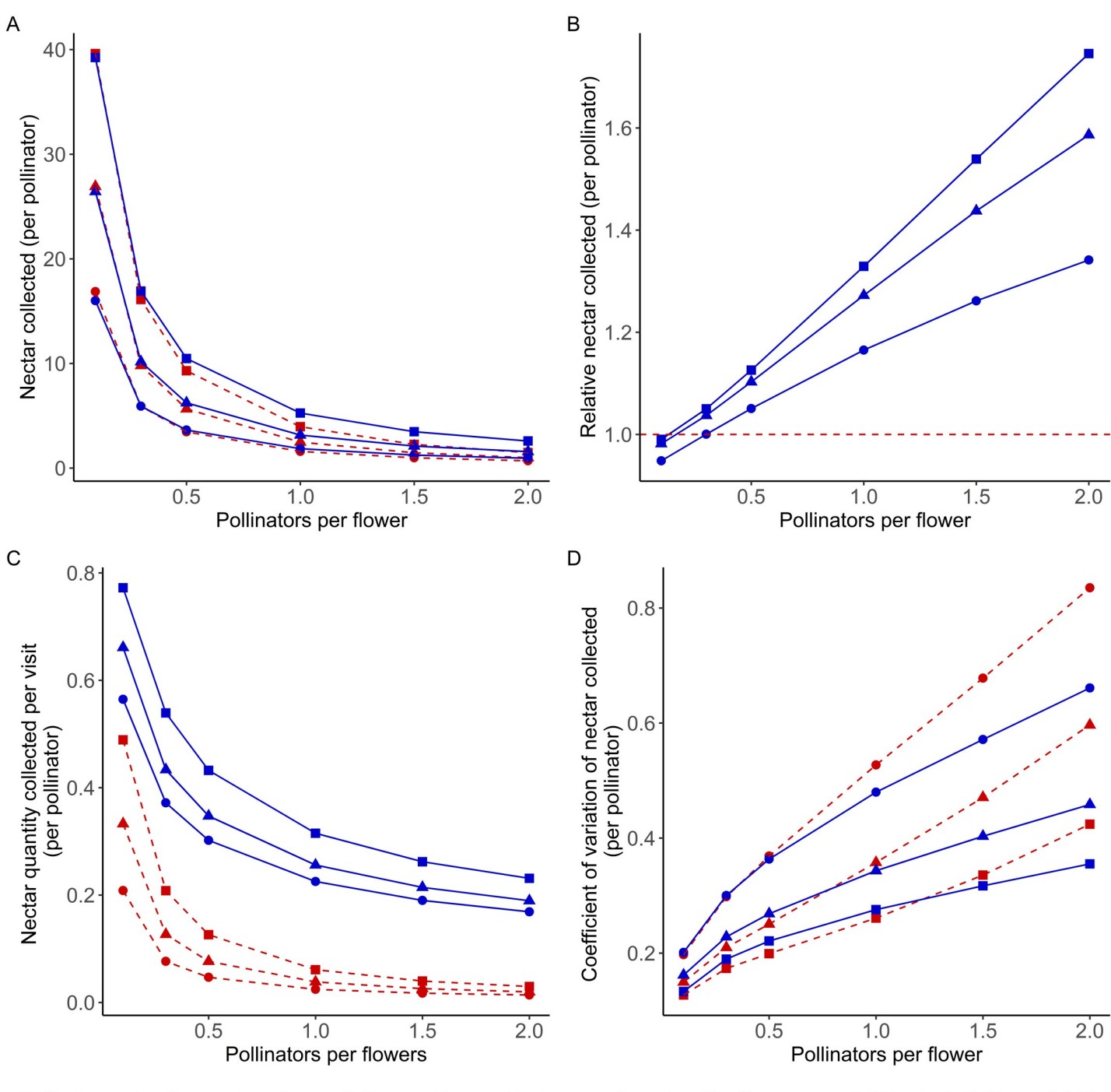

**Fig 4.** Pollinators foraging efficiency as a function of scent mark use and pollinators' relative abundance (here measured in relation to the number of flowers, i.e., pollinators per flower) (A) Mean amount of nectar collected by the pollinators during the simulation. (B) Ratio between the amount of nectar collected by pollinators using the scent mark in relation to that collected by pollinators not using the scent mark. The red dashed line represents the threshold where the amount of nectar collected is equal for both strategies. (C) Mean amount of nectar collected by pollinators at each flower visit. (D) Coefficient of variation of the mean amount of nectar collected by pollinators during the simulation. In each figure, the x-axis represents the relative abundance of pollinators compared to flowers.

We observed that the relative abundance of pollinators, which is directly related to the level of competition for nectar, strongly reduces the average amount of nectar harvested by pollinators, regardless of whether they use the scent mark (Fig 4A). The amount of nectar harvested increases with the level of nectar secretion. To better visualize the effect of using the scent mark for a given level of competition, we plotted the average amount of nectar collected by pollinators using the scent mark divided by the amount collected by those not using it (Fig 4B). This figure highlights the interaction between the effect of scent mark use and competition: when pollinator abundance is low, that is, with less than one pollinator for two flowers, scent mark use decreases the average amount of nectar collected. Nevertheless, for higher values of pollinator abundance, scent mark use results in the better efficiency of nectar collection, with the difference between the two strategies increasing with the level of competition.

Another aspect of the efficiency is to maximize the nectar quantity harvested per flower and avoid losing time visiting unrewarding flowers. As expected, the mean nectar quantity collected per visit is higher when using the scient mark, regardless of competition level or nectar refill coefficient (Fig 4C).

The use of the scent mark also affects the coefficient of variation of the nectar collected (Fig 4D). On the one hand, the coefficient of variation of the amount of nectar collected increases with the level of competition, and this increase is greater in individuals that do not use the scent mark: when pollinator abundance is low, that is, with less than one pollinator per flower, using the scent mark leads to higher coefficients of variation. On the other hand, for higher values of pollinator abundance, the use of scent marks results in less variability in the amount of nectar collected, with the difference between the two strategies increasing with the level of competition.

## 3.2 Amount of nectar in the flowers

We also examined the distribution of nectar within the flowers. Fig 5 shows the average amount of nectar remaining in the flowers at the end of the simulation and its coefficient of variation for the same nectar secretion values and relative abundances of pollinators as described above. As expected, the amount of nectar remaining in the flowers is lower when the abundance of pollinators increases (Fig 5A). The amount of nectar remaining at the end of the simulation is higher when pollinators use the scent mark than when they do not, with the ratio between the two increasing with the abundance of pollinators (Fig 5B).

Interestingly, when pollinators do not use the scent mark, the coefficient of variation of the remaining nectar increases sharply with the level of competition. On the contrary, when pollinators use the scent mark, the coefficient of variation is much lower and very similar for every value of competition and nectar coefficient (Fig 5C). When pollinators do not use the scent mark, competition increases the heterogeneity among the flowers, but when they use scent marks, this social information leads to a homogenization of the nectar distribution among the flowers.

## 3.3 Pollination service

The last aspect examined here was the pollination service, with an analysis of the number of visits per flower and the number of flowers potentially reached by pollen of each individual (Fig 6).

**3.3.1 Number of pollinator visits per flower.** Concerning the number of visits per flower, we can see that, as expected, when the abundance of pollinators increases, the number of flower visits also increases (Fig 6A). However, the number of visits is much lower when the pollinators use the scent mark. This number can be divided by 2 to 10 depending on the refill rate and the abundance of pollinators.

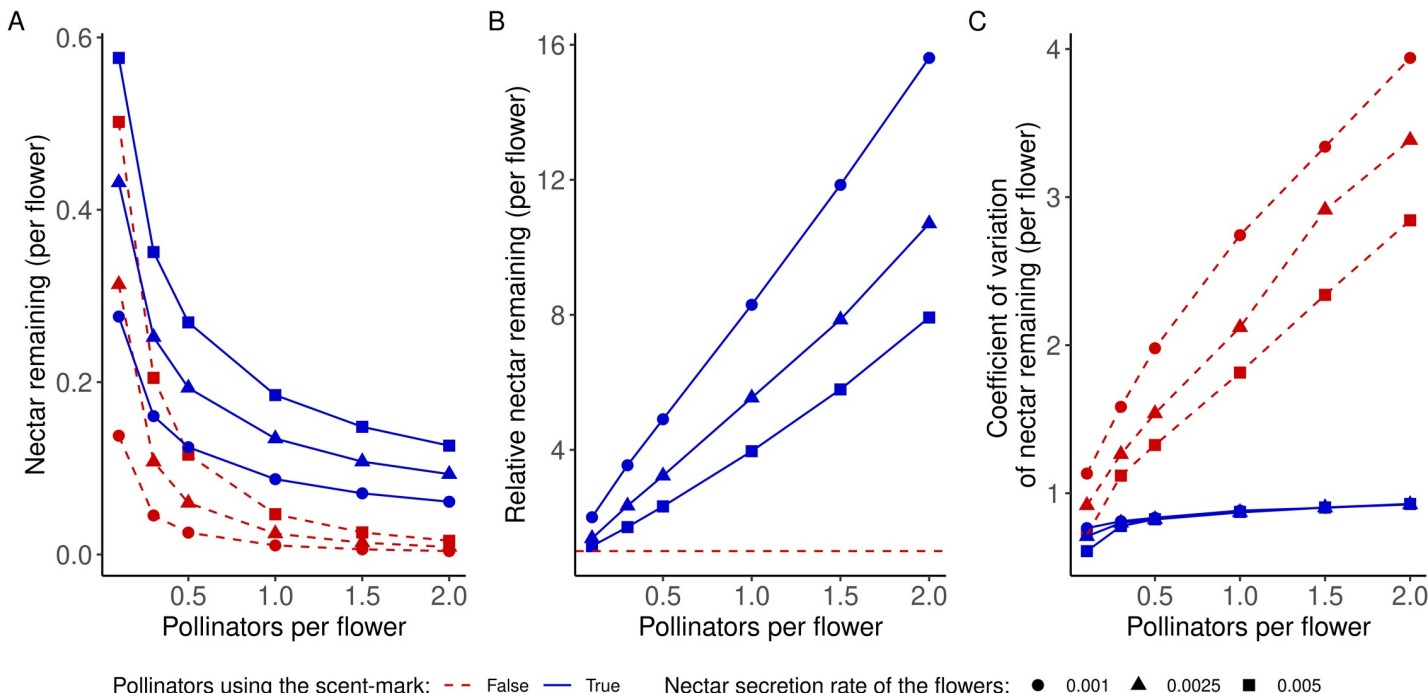

**Fig 5.** (A) Mean amount of nectar remaining in the flowers at the end of the simulation. (B) Ratio of the amount of nectar remaining in the flowers when pollinators do and do not use the scent mark. The red dashed line represents the threshold where the nectar quantity remaining is equal for both strategies. (C) Coefficient of variation of the nectar remaining in the flowers at the end of simulation. In each figure, the x-axis represents the relative abundance of pollinators compared to flowers.

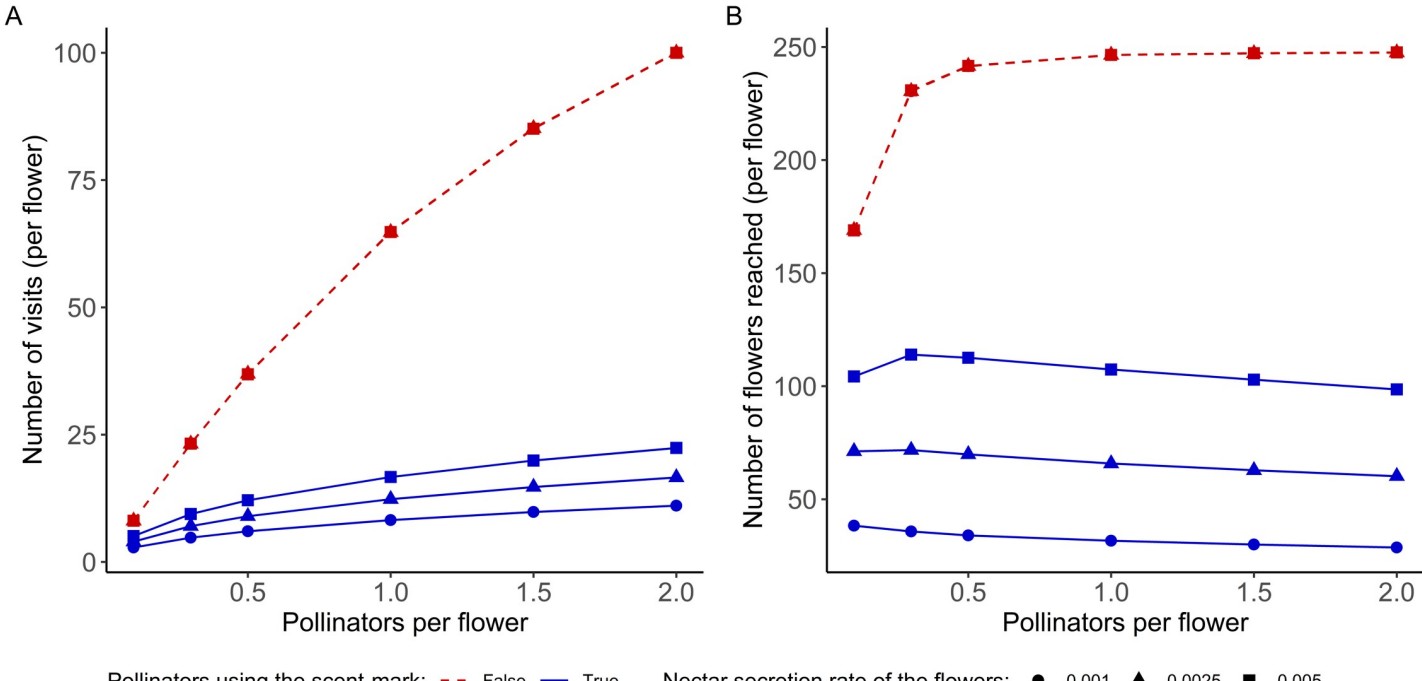

**Fig 6.** (A) Mean number of visits per flower during the simulation. (B) Mean number of flowers visited by a pollinator after he leaves the focal flower. In each figure, the x-axis represents the relative abundance of pollinators compared to flowers. The effect on the coefficient of variation can be found in the S3 File.

**3.3.2 Number of flowers reached by the pollen.** To complete our study of the pollination services, we analyzed how many flowers may be reached by the pollen of any given flower. Fig 6B presents the mean number of flowers visited by a pollinator after he leaves the focal flower. In line with the previous results, the pollen is less dispersed among the flower community when the pollinators used the scent-mark.

## 4. Discussion

Many insects can detect and recognize conspecifics by scent, an ability that can be found across taxonomic groups and ecological functions [31]. This ubiquitous ability to detect the scent of other insects and exploit odorant information is likely to have evolved before the existence of flowers [32]. In our agent-based model, pollinators can collect inadvertently provided cues in the form of repellent scent marks. Our simulations show that pollinators foraging in competitive contexts can collect more nectar when using scent marks as repellent cues, which is in accordance with experimental and observational studies [19, 33–35]. As expected, by using the scent mark to avoid recently visited flowers, pollinators no longer spend time in flowers with little or no nectar, leaving them more time to search elsewhere.

The relative advantage of using scent marks to avoid recently visited flowers depends on both the intensity of competition, which determines the average amount of nectar remaining in the flowers, and the relative cost of visiting depleted flowers. Pollinators face competition by exploitation, as their congeners deplete the floral resources. When the number of competitors is low, pollinators that do not use the scent mark may fare better than their "informed" counterparts, but as the relative abundance of competitors increases, the use of scent marks as repellent cues allows them to collect a greater amount of nectar.

In our model, the relative cost of visiting depleted flowers modulates the relationship between the intensity of competition and the advantage of using the scent mark, resulting in the relative advantage of foregoing the scent mark when the cost is low. Indeed, as the handling time of a given flower is likely to depend on its complexity, insects visiting simple flowers where nectar reward can be quickly assessed may forego the use of scent marks [32]. Saleh *et al.* [21] tested this hypothesis in a laboratory setting using bumblebees foraging on simple or complex artificial flowers. As predicted, bumblebees are more than twice as likely to reject complex scent-marked flowers compared to simple flowers. The handling time of a given flower is also likely to depend on the characteristics of the visiting insects. The co-evolution of pollinator feeding apparatus and flower traits results in visiting costs and rewards that may vary in relation to the visiting insect. Our model focused on scent mark use in homogeneous communities, but plant-pollinator communities are heterogeneous with networks characterized by varying degrees of both insect and flower generalism [36]. It is possible that the degree of generalism will affect the benefit of using repellent scent marks. We expect that the scent mark can be useful when foraging on generalist flowers but not when visiting flowers exploited by a small fraction of pollinators.

In our study, we were also interested in the effect of using repellent scent marks on the nectar resources. As we hypothesized, the use of social information correlates the behavior of pollinators and modifies the distribution of nectar resources. When the individuals use the scent mark as an inadvertent source of social information, the repartition of resources is more homogenous. Because the social information is a repellent, the correlation between visits will be negative, and flowers will not be overexploited. Moreover, because pollinators using scent marks no longer exploit flowers with little or no nectar, flowers should have enough time to produce a high amount of nectar between two visits. Our results corroborate this hypothesis, as the mean amount of nectar collected per flower is higher when pollinators use the scent-

mark. Consequently, the pollinators can detect a higher number of flowers, while the flowers are not underexploited, resulting in optimal use of nectar resources.

The homogenization of nectar resources is likely to affect the selective value of social information. Stout and Goulson [20] used a theoretical model to examine the consequences of nectar secretion rates on bumblebee's response to previously visited flowers. Their model predicted that the optimum frequency between visits was open to "cheating" by indiscriminate foragers that do not reject any flowers and would benefit from a homogeneous resource landscape when in low frequency. Likewise, we expect that the use of scent marks in our model is frequency-dependent and that at low frequencies, foragers foregoing the use of social information would fare better than their informed counterparts.

In addition to the homogenization of nectar resources, the use of repellent scent marks also impacts the number of visits per flower and the subsequent pollen dispersal among the community. Indeed, even if the overall number of flowers detected increases, the number of visits per flower decreases when pollinators use repellent scent marks. The relation between visitation rate and reproductive success will depends on a myriad of factors, including flower morphology, flower phenology, and pollinator behavior [37–39]. Our results suggest that the use of the scent-mark can decrease the pollination by decreasing the overall visitation rate, which has been shown to be correlate to pollen receipt [40], but also by limiting the pollen transfer to other flowers.

The mismatch between pollinator foraging success and pollination does not necessarily entail a deficient pollination service, particularly when we take diversity into account. Higher pollinator abundance and diversity have been shown to increase yield in pollinator-dependent crops in several independent studies [41]. More diverse communities should contain species with variable degrees of generalism, life span, and mobility, thus differing in their ability to use social information, which might increase pollination services and ensure resilience. The diversity in social information use among the pollinators' community is likely to lead to the co-existence of pollinators that use the scent-mark with some that do not. This is particularly likely when we consider possibility of frequency dependent selection of the scent-mark use [20] as discussed above.

To better understand how foraging strategies impact pollination services, it would be interesting to further explore the diversity of pollinators and plants and the associated foraging strategies. According to the optimal foraging theory, adaptation promotes pollinators' generalism when competition between pollinators is strong [42], which consequently facilitates plant diversity. Social foraging might further contribute to our understanding of the link between foraging strategies and diversity in plant-pollinator communities, as differential exploitation costs might maintain the diversity of pollinators.

## Supporting information

**S1 File. Detailed model description.** Model description is based on the ODD (Overview, Design concepts, Details) protocol.
(DOCX)

**S2 File. Sensitivity analysis.** Sensitivity analysis.
(DOCX)

**S3 File. Extra figures.** Coefficient of variation of the number of visits per flower during the simulation and the Coefficient of variation of the number of flowers which receive a visit of a pollinator after he leaves the focal one.
(DOCX)

## Acknowledgments

We thank Pau Capera-Aragones and an anonymous reviewer whose suggestions greatly improved the quality of this article.

## Author Contributions

**Conceptualization:** Elise Verrier, Emmanuelle Baudry, Carmen Bessa-Gomes.

**Formal analysis:** Elise Verrier.

**Funding acquisition:** Elise Verrier, Emmanuelle Baudry, Carmen Bessa-Gomes.

**Investigation:** Elise Verrier.

**Methodology:** Elise Verrier, Emmanuelle Baudry, Carmen Bessa-Gomes.

**Project administration:** Elise Verrier, Emmanuelle Baudry, Carmen Bessa-Gomes.

**Software:** Elise Verrier.

**Supervision:** Emmanuelle Baudry, Carmen Bessa-Gomes.

**Validation:** Elise Verrier, Emmanuelle Baudry, Carmen Bessa-Gomes.

**Visualization:** Elise Verrier.

**Writing – original draft:** Elise Verrier, Emmanuelle Baudry, Carmen Bessa-Gomes.

**Writing – review & editing:** Elise Verrier, Emmanuelle Baudry, Carmen Bessa-Gomes.

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
