## [Decision Letter · Decision Letter 0]

26 May 2021

PONE-D-21-10438

Modelling the effects of the repellent scent marks of pollinators on their foraging efficiency and the plant-pollinator community

PLOS ONE

Dear Dr. Verrier,

Thank you for submitting your manuscript to PLOS ONE. After careful consideration, we feel that it has merit but does not fully meet PLOS ONE’s publication criteria as it currently stands. Therefore, we invite you to submit a revised version of the manuscript that addresses the points raised during the review process.

Specifically, the reviewers raise important points both about the underlying assumptions of your models and the ecological implications of your results, and both of them need to be carefully addressed in a revised submission.

We look forward to receiving your revised manuscript.

Kind regards,

Ricardo Martinez-Garcia

Academic Editor

PLOS ONE

Journal Requirements:

Reviewers' comments:

Reviewer's Responses to Questions

**Comments to the Author**

1. Is the manuscript technically sound, and do the data support the conclusions?

Reviewer #1: Partly

Reviewer #2: Yes

2. Has the statistical analysis been performed appropriately and rigorously? 

Reviewer #1: Yes

Reviewer #2: Yes

3. Have the authors made all data underlying the findings in their manuscript fully available?

Reviewer #1: Yes

Reviewer #2: Yes

4. Is the manuscript presented in an intelligible fashion and written in standard English?

Reviewer #1: Yes

Reviewer #2: Yes

5. Review Comments to the Author

Reviewer #1: See attached file.......................................................................................... ............................................ ............................................ ............................................ ............................................

Reviewer #2: The paper studies the influence of repellent scent marks on the foraging efficiency in a group foraging setting. The authors provide an agent-based model, modeled after honey- and bumblebees, where recently visited flowers were less attractive to foragers due to the presence of a previously deposited repellent. The authors provide an excellent sensitivity analysis that highlight critical parameters that most strongly influence the evaluated metrics. By varying these parameters, they provide results on the effect of the level of competition on (group) foraging efficiency, remaining resources in the resource patches, and the total number of visits per patch.

Since foraging has an associated cost, i.e. handling time, the authors show that the use of repellent marks can increase foraging efficiencies in (highly) competitive systems. In contrast, repellent marks reduce the total number of visits per flower. As a result, the study has potential implications regarding pollinator systems (pollination service) that use the repellent scent marks.

While the results are very interesting, and the possible contrasting connection between high foraging efficiency and pollination services has potentially widespread implications, I believe the authors should:

- more strongly argue the reasons for specifications in their agent-based model (e.g. defend the choice of flower distribution, foraging cost, range of pollinators versus flowers, the random search, etc)

- provide more details on the effects the studied variables on the reported metrics (e.g. amount of nectar per flower visit, the impact of foraging cost [but see comment 3], etc )

- more thoroughly discuss the impact of their interesting results on pollination services

MAJOR COMMENTS:

---------------

1. The main goal of the authors is to display an apparent mismatch between optimal foraging strategies (the use of repellent scent marks) and the pollination service (number of visits per flower). However, other than illustrating this mismatch, I believe the study lacks a more thorough discussion on potential reasons and implications of this result. The authors only argue (in their Discussion) that this highlights the importance of taking diversity into account. With this, I am lead to believe that they most likely mean to state that different species of pollinators do not consider heterospecific scent marks. As a result, do the authors suggest that flowers that might contain a small amount of nectar are still visited by other pollinators? Or do the authors imply that pollination service can (potentially) be artificially controlled by modifying repellent scent marks? Otherwise, I see no advantage of decreased number of visits per flower for pollination service.

I believe the work would benefit from a more thorough discussion on the exact implications of their results, in light of the conclusion the authors aim to derive.

2. The authors report on foraging efficiencies, which is defined as the average amount of resources (nectar) consumed by each individual agent. In their random forest analysis, they provide influences of specific parameters, and conclude that the number of (discrete) steps highly increases the mean nectar quantity collected. However, the impact on the total amount of nectar collected obviously depends on the total time foraged; this is no surprise. Instead, I believe the authors should report on foraging efficiencies that consider nectar intake either per distance traveled, as is frequently done in foraging literature (see Viswanathan et al (1999), Bartumeus et al. (2005), among others), or per time unit. This way, the foraging efficiency becomes (approximately) independent of the total duration of the experiment, but this still depends strongly on the other measures such as the cost of foraging, regeneration rates, etc.

3. The authors mention that the time (cost) associated with flower exploitation depends strongly on flower complexity. They provide supplementary results on the effect of this cost on the total amount of nectar collected. Interestingly, systems that use repellent scent marks provide higher (albeit slightly) nectar intake than systems that do not use these marks. This effect appears to be largest when nectar regeneration rates are high (Fig. S2.2). Most interestingly, these results are obtained at a low number of pollinators (agents) per flower.

This highly interesting result already displays the benefits of repellent scent marks for (more) complex flower systems, but the authors decided to provide this information only in the supplementary material. I believe this result is critical, and provide more insight into the effect of repellent scent marks than the authors might think. Therefore, I believe that it should be moved (and should have an accompanying discussion) to the main body.

4. The authors discuss the effects of increased levels of competition, expressed by the number of available foraging sites compared to the number of foragers. However, I would argue that the system should not be inherently "competitive", since individual bees are part of a "collective" system. I agree with the authors that the collective as a whole would benefit from having the agents not spend much time in foraging sites with little resources. But the coefficient of variation loses its meaning in a collective context, as the goal of individuals is not to maximize their own resource intake, rather to aid the collective in maximizing the group foraging efficiency.

Therefore I believe that the coefficient of variation is not an informative metric to be used in this setting, unless it is used to explain possible increases in the (average) group efficiency. Unfortunately, other than mentioning the coefficient of variation, a more thorough explanation on why the coefficient of variation affects group foraging efficiencies is missing.

5. Similarly, the fact that the authors consider a competitive system reflected in the definition of foraging efficiency as the mean resource intake.

I would argue that the total amount of nectar (i.e. the sum over all individuals) is more appropriate in this setting, not the average intake. Do note that the average has much added value, but if, for example, repellent scents ensure a subset of the collective to forage much better, the collective as a whole can already benefit.

I believe it would therefore be appropriate if the authors more clearly argue why they study the particular metrics, and how they apply to (i) collective systems, or (ii) more competitive pollinator systems.

6. The authors study the amount of remaining resources (nectar) on the foraging sites (flowers) after the foraging task has ended. Surprisingly, the nectar remaining on the flowers is higher when agents use repellent scent marks. While the authors nicely identify that the variation in nectar remaining remains low in this case, it is unfortunate that the authors do not provide more details on this very interesting result. From Fig. 4 alone, I would conclude that the use of scent marks decreases the efficiency of resource usage, simply because more resources are left unused. However, from the Discussion (line 285-293), the authors mention "an optimization of nectar resources".

I am confused by this apparent discrepancy. Why is it precisely that the foraging is more efficient, yet the remaining amount of nectar in the flowers is higher?

7. The number of visits per flower for the different strategies is studied by the authors. Unsurprisingly, when repellent scent marks are absent, there is no change. This is reflected by the overlapping lines in Fig. 5A. Also unsurprisingly, the total number of flower visits is smaller in systems that do contain these repellents. However, while Fig. 3 displays an increase in the foraging efficiency, the authors do not provide data on the total resources (nectar) obtained at each flower visit. Obviously, for the foraging efficiency to increase, the amount of nectar obtained when a flower is visited must be higher. It would be nice if the authors can show that this is indeed the case, and perhaps link this with the previous comment on the remaining amount of nectar after foraging has finished.

Minor comments:

---------------

1. In their model, scent marks and nectar quantity are updated linearly. However, the authors provide no sources for this type of growth. Furthermore, while the authors assume that foraging on the flowers is probabilistic, it has been shown that it is more likely to be a binary decision, i.e. after a fixed time length, the flower will be visited again (see e.g. Stout, Goulson (2001), Nauta (Int. Conf. on Swarm Intelligence 2020), Hrncir et al. (2004)). While the effect of this will most likely be minimal, it would be beneficial to the quality of the work if the authors could discuss their choice of linear growth or decay in the light of existing works.

2. The random search of the agents is defined by a change in travel angle, sampled from a Gaussian. The authors perform a sensitivity analysis, and choose a specific value for the standard deviation, while the mean is 0. While the likelihood of sampling changes in travel angles more than 2π radians is slim, this is still possible due to there being no truncation. Moreover, the model resembles the Correlated Random Walk (CRW), which has been extensively studied to model random searches (see e.g. Bovet, Benhamou (1988), Bartumeus et al. (2005), Nauta et al. (Royal Society Interface 2020)).

While I would argue that the effect on the results would be minimal due to the choice of standard deviation, I believe the authors should more strongly defend their choice of the random search, or consider (slightly) more realistic random walks such as the CRW.

3. Pertaining to the choices of pollinators per flower, I believe it would be beneficial if the authors could report on observed values in natural systems. If, for example, the number of pollinators per flower is observed to be much smaller than 1, then the use of repellent signals decreases foraging efficiency, and the main hypothesis of the paper should be revised. Therefore, I would advise the authors to strengthen their results with empirical observations (if available), which further increases the implication of the work.

4. In the Discussion, the authors devote a paragraph to discussing generalism and learning. They conclude that pollination services can potentially be increased by diverse pollinator communities that express more heterogeneity. However, this paragraph feels out of place, as they have not discussed learning in any other context in their study. While learning obviously has its place in foraging, I believe the authors should provide more details on the link between learning, generalism and their hypothesis that more diverse pollinator communities can provide better pollination services.

5. The authors report on the precise implementation details in the main body. These details are better suited to be included (and they are!), and should not be included in the main body. Additionally, while it is most likely a subjective matter, I believe citing the Python language and (extremely) widely used libraries is not necessary, especially not within the main body of the text. If the authors desire, I believe it is no problem to include them in supplementary material.

6. While the quality of the language throughout the document is high, while still remaining very clear, some sentences (or structures) slightly suffer from repetition or strange word choice.

Examples of these sentences (and possibly, suggested changes):

line 50-58: First two sentences are essentially repeated by the last two.

line 97: "environmental space"  "environment"

line 114: "foraging journey"  "foraging duration"

line 115: "toroid meadow"  "environment"

Minor typos:

------------

line 164: "Sensibility" -> "Sensitivity"

line 287: "sent" -> "scent"

Suppl. material, line 30: "... for our problematic."  "... for our problem."

Figures:

In general, I believe that the figures will benefit from inclusion of a legend, instead of providing figure details in the caption. Additionally, perhaps different linestyles (and filled/unfilled markers) can provide a more clear distinction between the different curves. Finally, but this is perhaps subjective, explicitly mentioning what the x-axis represent is often unnecessary as it is clear from both the text and the x-label.

Fig. S2.2 appears to times. The figure on the effect of cost should be relabeled to Fig. S2.3.

REFERENCES:

-----------

Viswanathan GM, Buldyrev SV, Havlin S, Da Luz M, Raposo E, Stanley HE. (1999), Optimizing the success of random searches. Nature 401(6756):911. https://doi.org/10.1038/44831

Hybrid foraging in patchy environments using spatial memoryJ Nauta, Y Khaluf, P Simoens Journal of the Royal Society Interface 17 (166), 20200026

Bartumeus, F., da Luz, M.G..E., Viswanathan, G.M. and Catalan, J. (2005), ANIMAL SEARCH STRATEGIES: A QUANTITATIVE RANDOM‐WALK ANALYSIS. Ecology, 86: 3078-3087. https://doi.org/10.1890/04-1806

Memory Induced Aggregation in Collective ForagingJ Nauta, P Simoens, Y Khaluf International Conference on Swarm Intelligence, 176-189

Bovet, P., Benhamou, S. (1988), Spatial analysis of animals' movements using a correlated random walk model, Journal of Theoretical Biology, 131-4: 419-433. https://doi.org/10.1016/S0022-5193(88)80038-9

Stout, J.C., Goulson, D. (2001), The use of conspecific and interspecific scent marks by foraging bumblebees and honeybees. Animal Behavior, 62-1: 183,189. https://doi.org/10.1006/anbe.2001.1729

Hrncir, M. Jarau S., Zucchi, R., Barth, F.G. (2004), On the origin and properties of scent marks deposited at the food source by a stingless bee, Melipona seminigra. Apidologie, 35-1: 3-13. https://doi.org/10.1051/apido:2003069

6. PLOS authors have the option to publish the peer review history of their article (what does this mean?). If published, this will include your full peer review and any attached files.

Reviewer #1: No

Reviewer #2: No

---

## [Author Response · Author response to Decision Letter 0]

17 Jul 2021

All the information below are in the document entitled Response to the Reviewers :

Dear Editor,

We thank you for considering our manuscript, and we are grateful to both reviewers for their helpful and constructive comments, which substantially improved the content and clarity of our manuscript. Below, we provide a point-by-point response to all comments. Where needed, we quoted the revised text from our manuscript, referring it by its line numbers (revised version). We addressed all comments in detail. 

The main change of this version is, following Reviewer #1 suggestion, we now present results regarding the number of flowers that are likely to receive the pollen of any given focal flower; this new measure of pollen transfer improves our discussion on the pollination service provided by insects in relation to their behavioral strategy.

Sincerely yours,

On behalf of all authors,

Elise Verrier

Reviewer #1 Comments

Overview: In their manuscript, the authors explore the use of scent marks left on the corolla of flowers by pollinators, as a cue to avoid recently depleted resources. They use a spatially explicit agent-based model to examine the effects at the individual and collective level. They show that, at the individual level, the use of scent marks increases foraging efficiency (increases nectar collection) when competition for resource is important. At the community level, this resulted in a homogenization of the resource depleted. Their results also show that the use of scent marks decreased the number of flower visits, which can potentially reduce the pollination services. 

The manuscript is written clearly and the results are intuitive and appear to be in line with empirical data. My main concern is about their conclusion saying that the use of scent marks can potentially decrease pollination services.

Major comments:

1. In their results, the authors show that the use of scent marks reduces the number of flower visits. They mention many times that this reduction in flower visits implies a reduction of pollination services. This implication is not necessarily true, especially because their results also show that the use of scent marks homogenizes the visits among flowers. In line 307, the authors cite [40] which indicates that the visitation rate is related to pollen receipt. We can understand that pollen receipt is related to pollination services although no reference is provided here. However, the distribution of the pollen among flowers and not only the total amount is important. To put an example, it is not clear to me that a landscape consisting of 5 flowers with 100 visits to one of the flowers and 0 visits to all the others, has more pollination services than a landscape in which each of the 5 flowers get 10 visits. In the first case, the total number of visits is 100, in the second case is 50, but I expect the second case to have higher pollination services. This example is very related to their case of study and so their conclusion “fewer flowers’ visits imply less pollination” isn’t generally valid. The decrease of flower visits when using scent marks supports the reduction of pollination services, but the decrease of the coefficient of variation of the number of visits per flower when using scent marks supports the increase of pollination services. For that reason, the authors should provide a better justification for their conclusion or change it. To evaluate the effects of scent marks on pollination services another metric should be used. 

Thank you very much for pointing this aspect. We add a new metric as you recommend, the number of flowers reached by pollen, lines 263-267 in the results and lines 316-323in the discussion. This new metric indicates that pollen transfer is affected by the behavioral strategy, the use of the scent mark resulting in a transfer to fewer flowers than otherwise. 

I suggest plotting the number of flowers with no visits, or the number of flowers with less than “x” visits as a possible indicator, although I think a conclusive argument would be hard to achieve.

Although the number of visits each flower receives varies with pollinators’ behavior, almost all flowers receive at least one visit. In the worse scenario, when the number of pollinators is very low, only two flowers over 250 received zero visits (see figure inserted below). We believe that the pollen transfer measure presented above is more informative than the number of flowers receiving zero visits. 

Figure: number of flowers which have received no visit during the simulation as a function of pollinators’ relative abundance. 

2. The authors do not specify with enough detail the type of movement their model bees do. In Fig.2, the sentence “Move of one arbitrary unit in almost the same direction as previously” is too vague. What does an arbitrary unit mean? What does almost the same direction mean?

Thank for the comment, we change the figure in the supplementary 1, see below. 

3. The authors do not mention previous modelling work about scent marks. For example, there is quite a bit made by Mark Lewis and other authors in papers such: Brian K Briscoe, Mark A Lewis, and Stephen E Parrish. Home range formation in wolves due to scent marking. Bulletin of Mathematical Biology, 64(2):261–284, 2002.-

Thank you for the reference. We have now included this reference in the introduction (lines 40-42).

Minor Comments:

1. 49 Replace “their consumption” with “the consumption” 

-> Thank you

2. 51 Can you include some examples illustrating what is meant by “personal information”? Does personal information also indicate memory? 

-> Personal information is related to individuals experience of their environment and can indeed entail memory. We have now added an example (lines 51-53). 

3. 85-89. The sentences “We aim to ... nectar available on the flower” says the same as the next lines “we record ... and mean number of visits per flower”, it sounds repetitive. 

-> We have re-writen the introductory paragraph of the “Material & methods section” to avoid this repetition (lines 85-95).

4. 164 Section number “2.3” is repeated. 

-> Thank you, we have corrected this.

Plots: 

1. Use the same notation on the vertical axes in Figs. 3, 4 and 5 when possible. For example, in Fig. 3, plot A has on the axis “Nectar collected”, so plot B should therefore be “Relative nectar collected”. In Fig. 4, plot A has on the axis “Nectar remaining” (is it per flower? if so, then put it, if it is not, then don’t put per flower in plot B, since in B is relative abundance, it doesn’t matter if it is per flower or not). In Fig. 5, plot A says “Nectar remaining”, plot B says “Relative abundance of nectar per flower” shouldn’t it be simply “Relative nectar remaining”? 

-> Thank you for the suggestions. We have renamed the axis of figures 4, 5 and 6 accordingly. 

2. There is no dotted red line in Fig. 3 as mentioned in the caption of the figure. 

-> Thank you, this has been corrected.

3. Make a legend for Figs. 3, 4 and 5. It is much more visual than going to the caption every time to see which colour corresponds to the use of scent marks and which not. The legend should also include information about the nectar replenishment rates represented by “squares”, “triangles” and “circles”. 

-> Thank you

4. I also encourage you to use a different colour than “red” (only for scent mark pollinators) and “blue” (only for no-scent mark pollinators) for the case of relative abundances. 

-> We decided to keep the red color on those figures. In fact, the red dashed line is set at one, but also represents the focal value of the no-use scent mark pollinator / the focal value of the no-use scent mark pollinator. (The blue reflects: the focal value of the use scent mark pollinator / the focal value of the no-use scent mark pollinator.) As we changed the legend and the linetype for the other graphics, this is hopefully clearer. 

Suggestions: 

1. I suggest adding a future direction considering the memory of scent marks (so foragers would do flights more directly to their favourite locations in the landscape) or scent mark dependent dynamics (foragers moving faster when they perceive high concentrations of scent marks around them). These types of behaviours can potentially increase the efficiency of foragers using scent marks even more.

Thank you for your suggestion. It is indeed a direction that we plan to explore in the future. We are very interested in the interplay between personal and social information, and information on resource use by other pollinators may be an important source of information on resource value. We are currently working on it.

2. If your model can reproduce the behaviour stated in lines 272-274, why you don’t add something in the abstract? It seems relevant to me.

This has been included in the abstract, were we state, “We showed that the use of scent marks as a source of information significantly increased the foraging efficiency of pollinators except when competition between pollinators was very low” (lines 27-29).

 

Reviewer #2: The paper studies the influence of repellent scent marks on the foraging efficiency in a group foraging setting. The authors provide an agent-based model, modeled after honey- and bumblebees, where recently visited flowers were less attractive to foragers due to the presence of a previously deposited repellent. The authors provide an excellent sensitivity analysis that highlight critical parameters that most strongly influence the evaluated metrics. By varying these parameters, they provide results on the effect of the level of competition on (group) foraging efficiency, remaining resources in the resource patches, and the total number of visits per patch.

 Since foraging has an associated cost, i.e. handling time, the authors show that the use of repellent marks can increase foraging efficiencies in (highly) competitive systems. In contrast, repellent marks reduce the total number of visits per flower. As a result, the study has potential implications regarding pollinator systems (pollination service) that use the repellent scent marks.

While the results are very interesting, and the possible contrasting connection between high foraging efficiency and pollination services has potentially widespread implications, I believe the authors should:

 - more strongly argue the reasons for specifications in their agent-based model (e.g. defend the choice of flower distribution, foraging cost, range of pollinators versus flowers, the random search, etc)

 - provide more details on the effects the studied variables on the reported metrics (e.g. amount of nectar per flower visit, the impact of foraging cost [but see comment 3], etc )

 - more thoroughly discuss the impact of their interesting results on pollination services

MAJOR COMMENTS:

1. The main goal of the authors is to display an apparent mismatch between optimal foraging strategies (the use of repellent scent marks) and the pollination service (number of visits per flower). However, other than illustrating this mismatch, I believe the study lacks a more thorough discussion on potential reasons and implications of this result. The authors only argue (in their Discussion) that this highlights the importance of taking diversity into account. With this, I am lead to believe that they most likely mean to state that different species of pollinators do not consider heterospecific scent marks. As a result, do the authors suggest that flowers that might contain a small amount of nectar are still visited by other pollinators? Or do the authors imply that pollination service can (potentially) be artificially controlled by modifying repellent scent marks? Otherwise, I see no advantage of decreased number of visits per flower for pollination service. I believe the work would benefit from a more thorough discussion on the exact implications of their results, in light of the conclusion the authors aim to derive.

-> Thank you for the comment, following your advice we re-write and detail this part of the discussion, see 324-332. 

2. The authors report on foraging efficiencies, which is defined as the average amount of resources (nectar) consumed by each individual agent. In their random forest analysis, they provide influences of specific parameters, and conclude that the number of (discrete) steps highly increases the mean nectar quantity collected. However, the impact on the total amount of nectar collected obviously depends on the total time foraged; this is no surprise. Instead, I believe the authors should report on foraging efficiencies that consider nectar intake either per distance traveled, as is frequently done in foraging literature (see Viswanathan et al (1999), Bartumeus et al. (2005), among others), or per time unit. This way, the foraging efficiency becomes (approximately) independent of the total duration of the experiment, but this still depends strongly on the other measures such as the cost of foraging, regeneration rates, etc. 

-> Thank you for highlighting this point. In fact, the number of steps changes only during the sensitivity analyses, for the rest of the study all simulations have an identical number of steps (1000 steps). So, we could consider the nectar quantity collected per unit of time, but the results would be not affected by this new measure. 

3. The authors mention that the time (cost) associated with flower exploitation depends strongly on flower complexity. They provide supplementary results on the effect of this cost on the total amount of nectar collected. Interestingly, systems that use repellent scent marks provide higher (albeit slightly) nectar intake than systems that do not use these marks. This effect appears to be largest when nectar regeneration rates are high (Fig. S2.2). Most interestingly, these results are obtained at a low number of pollinators (agents) per flower. This highly interesting result already displays the benefits of repellent scent marks for (more) complex flower systems, but the authors decided to provide this information only in the supplementary material. I believe this result is critical, and provide more insight into the effect of repellent scent marks than the authors might think. Therefore, I believe that it should be moved (and should have an accompanying discussion) to the main body. 

-> We agree with the reviewer, and we now present this result in the main body of the manuscript. The results are now presented in section 3.1.1 (Impact of the cost, lines: 185-195; 284-289)

4. The authors discuss the effects of increased levels of competition, expressed by the number of available foraging sites compared to the number of foragers. However, I would argue that the system should not be inherently "competitive", since individual bees are part of a "collective" system. I agree with the authors that the collective as a whole would benefit from having the agents not spend much time in foraging sites with little resources. But the coefficient of variation loses its meaning in a collective context, as the goal of individuals is not to maximize their own resource intake, rather to aid the collective in maximizing the group foraging efficiency. Therefore, I believe that the coefficient of variation is not an informative metric to be used in this setting, unless it is used to explain possible increases in the (average) group efficiency. Unfortunately, other than mentioning the coefficient of variation, a more thorough explanation on why the coefficient of variation affects group foraging efficiencies is missing. 

-> Thank you for raising this point. Indeed, our study does not take into account the “collective” dimension. This is because our model is centered on a community of pollinating insects exploiting a flower meadow and not on one colony of social insects. Among pollinators who use the mark, there are social species living in colonies such as honeybees and bumblebees (which represents a large number of species, several dozen species for example in Europe for bumblebees alone), but also solitary species (Yokoi and Fujisaki, 2009). If we consider the pollinating insects using a scent mark that forage in a flower meadow, it seems likely that the majority of them belong to different species, and that the social dimension is in the minority compared to the competition dimension. We changed the sentence lines 61-64 in the introduction to underline that both social and solitary species can use scent marks.”.

5. Similarly, the fact that the authors consider a competitive system reflected in the definition of foraging efficiency as the mean resource intake. I would argue that the total amount of nectar (i.e. the sum over all individuals) is more appropriate in this setting, not the average intake. Do note that the average has much added value, but if, for example, repellent scents ensure a subset of the collective to forage much better, the collective as a whole can already benefit. I believe it would therefore be appropriate if the authors more clearly argue why they study the particular metrics, and how they apply to (i) collective systems, or (ii) more competitive pollinator systems. 

Thank you for your comment. It will be clearly an analyse that must be addressed if we add a social aspect and in consequence a collective system like a hive in our study. However, in a competition context only, we think that the estimate per individual is more pertinent, as it allows us to compare across variable pollinators abundance.

5. The authors study the amount of remaining resources (nectar) on the foraging sites (flowers) after the foraging task has ended. Surprisingly, the nectar remaining on the flowers is higher when agents use repellent scent marks. While the authors nicely identify that the variation in nectar remaining remains low in this case, it is unfortunate that the authors do not provide more details on this very interesting result. From Fig. 4 alone, I would conclude that the use of scent marks decreases the efficiency of resource usage, simply because more resources are left unused. However, from the Discussion (line 285-293), the authors mention "an optimization of nectar resources". I am confused by his apparent discrepancy. Why is it precisely that the foraging is more efficient, yet the remaining amount of nectar in the flowers is higher? 

-> Thank you for your remark. As you suggested in the next comment, we added a new figure (fig4 -line 201) and details the discussion line 298-307.

7. The number of visits per flower for the different strategies is studied by the authors. Unsurprisingly, when repellent scent marks are absent, there is no change. This is reflected by the overlapping lines in Fig. 5A. Also unsurprisingly, the total number of flower visits is smaller in systems that do contain these repellents. However, while Fig. 3 displays an increase in the foraging efficiency, the authors do not provide data on the total resources (nectar) obtained at each flower visit. Obviously, for the foraging efficiency to increase, the amount of nectar obtained when a flower is visited must be higher. It would be nice if the authors can show that this is indeed the case, and perhaps link this with the previous comment on the remaining amount of nectar after foraging has finished. 

-> Thank you for your relevant suggestion. We added this new analysis of the efficiency of the nectar collection by pollinators to our results (lines 298-307). They are illustrated by fig 4 (line 201). 

Minor comments:

---------------

 1. In their model, scent marks and nectar quantity are updated linearly. However, the authors provide no sources for this type of growth. 

-> Thank you for highlighting this point. There are not many studies about nectar refill in flowers, but we base our hypothesis on the work of Stout and Goulson (Stout and Goulson, 2002). They tested 4 species, and three of them had a linear refill of nectar (Lotus corniculatus; Melilotus officinalis; Phaceliatanacetifolia). Nonetheless, it was a mistake to do not cite them in the materials and methods, we changed that line 145. 

Furthermore, while the authors assume that foraging on the flowers is probabilistic, it has been shown that it is more likely to be a binary decision, i.e. after a fixed time length, the flower will be visited again (see e.g. Stout, Goulson (2001), Nauta (Int. Conf. on Swarm Intelligence 2020), Hrncir et al. (2004)). While the effect of this will most likely be minimal, it would be beneficial to the quality of the work if the authors could discuss their choice of linear growth or decay in the light of existing works. 

-> We agree with the reviewer that this is an aspect that deserves further discussion. We had this discussion ourselves during the construction of the model. We wanted to keep the model as simple as possible and limit the number of parameters, but we wished to allow for individual variation. We considered that transforming this binary phenomenon to a stochastic process, we allow for a specific threshold for each pollinator-flower interaction. This is now discussed in the supplementary material 1 section (99-100). 

2. The random search of the agents is defined by a change in travel angle, sampled from a Gaussian. The authors perform a sensitivity analysis, and choose a specific value for the standard deviation, while the mean is 0. While the likelihood of sampling changes in travel angles more than 2π radians is slim, this is still possible due to there being no truncation. Moreover, the model resembles the Correlated Random Walk (CRW), which has been extensively studied to model random searches (see e.g. Bovet, Benhamou (1988), Bartumeus et al. (2005), Nauta et al. (Royal Society Interface 2020)). While I would argue that the effect on the results would be minimal due to the choice of standard deviation, I believe the authors should more strongly defend their choice of the random search, or consider (slightly) more realistic random walks such as the CRW.

We agree with the reviewers that CRW would have been the optimal choice for the present analysis. The current model was developed as a modelling framework to examine a set of questions and we wished to model movement following other criteria as well, as in previous models by one the authors (Deygout et al., 2009, 2010). We believe that the present system, when taking the deviation from a gaussian approaches the properties of the CRW. We have now revised the model description to acknowledge it. The risk of travel angles more than 2π radians is discussed in the sensibility analysis.

3. Pertaining to the choices of pollinators per flower, I believe it would be beneficial if the authors could report on observed values in natural systems. If, for example, the number of pollinators per flower is observed to be much smaller than 1, then the use of repellent signals decreases foraging efficiency, and the main hypothesis of the paper should be revised. Therefore, I would advise the authors to strengthen their results with empirical observations (if available), which further increases the implication of the work. 

-> Thank you for your advice. In fact, empirical data on the subject are scarce. Most of the results we had were from experiments in lab or in greenhouse, where they try to avoid competition. In the lab, we can cite the recent work of Cervantes‐Loreto et al. (2021), which use between 4 and 16 bumblebees for 32 artificial flowers. The metric widely used in greenhouse studies is the number of individuals per ha without any precision about the number of flowers. Some studies are more precise and give indication of the number of worker and plants, between 70-90 bumblebee workers for 20 tomatoes plants (Vergara and Fonseca-Buendía, 2012) For the melon cultivate in green house, the optimal number of honeybees for 100 plants (with between 15 and 60 flowers per plants) is 5000-6000 individuals (Filip Slavkovic 2021, personal communications). And finally, this is also linked with the specie used, for example Zhang et al, set a colony of 6000 apis mellifera workers for 45 peach trees and only 60 bumblebee workers for the same number of trees. Having empirical data to adjust our model will drastically improve our study, we sincerely hope that in the future we will have the opportunity to confront our model with the reality. 

4. In the Discussion, the authors devote a paragraph to discussing generalism and learning. They conclude that pollination services can potentially be increased by diverse pollinator communities that express more heterogeneity. However, this paragraph feels out of place, as they have not discussed learning in any other context in their study. While learning obviously has its place in foraging, I believe the authors should provide more details on the link between learning, generalism and their hypothesis that more diverse pollinator communities can provide better pollination services.

We followed the reviewers comment and excluded the paragraph discussing Jones and Agrawal, (2017) work on the relation between generalism and learning.

5. The authors report on the precise implementation details in the main body. These details are better suited to be included (and they are!) and should not be included in the main body. Additionally, while it is most likely a subjective matter, I believe citing the Python language and (extremely) widely used libraries is not necessary, especially not within the main body of the text. If the authors desire, I believe it is no problem to include them in supplementary material. 

The citation of the library has been moved to the supplementary 1. 

6. While the quality of the language throughout the document is high, while still remaining very clear, some sentences (or structures) slightly suffer from repetition or strange word choice.

 Examples of these sentences (and possibly, suggested changes):

 line 50-58: First two sentences are essentially repeated by the last two.

 line 97: "environmental space"  "environment" -> Thank you

line 114: "foraging journey"  "foraging duration" -> Thank you

line 115: "toroid meadow"  "environment" -> Thank you

Minor typos:

 ------------

 line 164: "Sensibility" -> "Sensitivity" -> Thank you

line 287: "sent" -> "scent" -> Thank you

Suppl. material, line 30: "... for our problematic."  "... for our problem." -> Thank you

Figures:

 In general, I believe that the figures will benefit from inclusion of a legend, instead of providing figure details in the caption. 

-> Thank you, it has been changed. 

Additionally, perhaps different linestyles (and filled/unfilled markers) can provide a more clear distinction between the different curves. 

-> We decided to have different lintypes for the use of the scent-mark, but we keep all symbols filled, which seem easier to follow in our point of view, especially for the figure 4 (ex- figure 3). 

Finally, but this is perhaps subjective, explicitly mentioning what the x-axis represent is often unnecessary as it is clear from both the text and the x-label.

Fig. S2.2 appears to times. The figure on the effect of cost should be relabeled to Fig. S2.3. 

-> The figure has been moved to the main manuscript.

REFERENCES:

 -----------

 Viswanathan GM, Buldyrev SV, Havlin S, Da Luz M, Raposo E, Stanley HE. (1999), Optimizing the success of random searches. Nature 401(6756):911. https://doi.org/10.1038/44831

Hybrid foraging in patchy environments using spatial memoryJ Nauta, Y Khaluf, P Simoens Journal of the Royal Society Interface 17 (166), 20200026

Bartumeus, F., da Luz, M.G..E., Viswanathan, G.M. and Catalan, J. (2005), ANIMAL SEARCH STRATEGIES: A QUANTITATIVE RANDOM‐WALK ANALYSIS. Ecology, 86: 3078-3087. https://doi.org/10.1890/04-1806

Memory Induced Aggregation in Collective ForagingJ Nauta, P Simoens, Y Khaluf International Conference on Swarm Intelligence, 176-189

Bovet, P., Benhamou, S. (1988), Spatial analysis of animals' movements using a correlated random walk model, Journal of Theoretical Biology, 131-4: 419-433. https://doi.org/10.1016/S0022-5193(88)80038-9

Stout, J.C., Goulson, D. (2001), The use of conspecific and interspecific scent marks by foraging bumblebees and honeybees. Animal Behavior, 62-1: 183,189. https://doi.org/10.1006/anbe.2001.1729

Hrncir, M. Jarau S., Zucchi, R., Barth, F.G. (2004), On the origin and properties of scent marks deposited at the food source by a stingless bee, Melipona seminigra. Apidologie, 35-1: 3-13. https://doi.org/10.1051/apido:2003069

Bibliography: 

Cervantes‐Loreto, A. et al. (2021) ‘The context dependency of pollinator interference: How environmental conditions and co‐foraging species impact floral visitation’, Ecology Letters. Edited by R. Irwin, 24(7), pp. 1443–1454. doi: 10.1111/ele.13765.

Deygout, C. et al. (2009) ‘Modeling the impact of feeding stations on vulture scavenging service efficiency’, Ecological Modelling, 220(15), pp. 1826–1835. doi: 10.1016/j.ecolmodel.2009.04.030.

Deygout, C. et al. (2010) ‘Impact of food predictability on social facilitation by foraging scavengers’, Behavioral Ecology, 21(6), pp. 1131–1139. doi: 10.1093/beheco/arq120.

Jones, P. L. and Agrawal, A. A. (2017) ‘Learning in Insect Pollinators and Herbivores’, Annual review of entomology, 62, pp. 53–71.

Stout, J. and Goulson, D. (2002) ‘The influence of nectar secretion rates on the responses of bumblebees ( Bombus spp.) to previously visited flowers’, Behavioral Ecology and Sociobiology, 52(3), pp. 239–246. doi: 10.1007/s00265-002-0510-2.

Vergara, C. H. and Fonseca-Buendía, P. (2012) ‘Pollination of Greenhouse tomatoes by the Mexican bumblebee Bombus ephippiatus (Hymenoptera: Apidae)’, Journal of Pollination Ecology, 7, pp. 27–30. doi: 10.26786/1920-7603(2012)1.

Yokoi, T. and Fujisaki, K. (2009) ‘Recognition of scent marks in solitary bees to avoid previously visited flowers’, Ecological Research, 24(4), pp. 803–809. doi: 10.1007/s11284-008-0551-8.

---

## [Decision Letter · Decision Letter 1]

19 Aug 2021

Modelling the effects of the repellent scent marks of pollinators on their foraging efficiency and the plant-pollinator community

PONE-D-21-10438R1

Dear Dr. Verrier,

We’re pleased to inform you that your manuscript has been judged scientifically suitable for publication and will be formally accepted for publication once it meets all outstanding technical requirements.

Kind regards,

Ricardo Martinez-Garcia

Academic Editor

PLOS ONE

Additional Editor Comments (optional):

Reviewers' comments:

Reviewer's Responses to Questions

**Comments to the Author**

1. If the authors have adequately addressed your comments raised in a previous round of review and you feel that this manuscript is now acceptable for publication, you may indicate that here to bypass the “Comments to the Author” section, enter your conflict of interest statement in the “Confidential to Editor” section, and submit your "Accept" recommendation.

Reviewer #1: All comments have been addressed

Reviewer #2: All comments have been addressed

2. Is the manuscript technically sound, and do the data support the conclusions?

Reviewer #1: Yes

Reviewer #2: Yes

3. Has the statistical analysis been performed appropriately and rigorously? 

Reviewer #1: N/A

Reviewer #2: Yes

4. Have the authors made all data underlying the findings in their manuscript fully available?

Reviewer #1: Yes

Reviewer #2: Yes

5. Is the manuscript presented in an intelligible fashion and written in standard English?

Reviewer #1: Yes

Reviewer #2: Yes

6. Review Comments to the Author

Reviewer #1: Overall, the authors addressed the last comments appropriately. My only last minor concern is regarding the meaning of the word "view_radius" in Fig.S2.2, which I think is not intuitive nor described.

Reviewer #2: (No Response)

7. PLOS authors have the option to publish the peer review history of their article (what does this mean?). If published, this will include your full peer review and any attached files.

Reviewer #1: **Yes: **Pau Capera-Aragones

Reviewer #2: No

---

## [Editor Report · Acceptance letter]

27 Aug 2021

PONE-D-21-10438R1 

Modelling the effects of the repellent scent marks of pollinators on their foraging efficiency and the plant-pollinator community  

Dear Dr. Verrier:

I'm pleased to inform you that your manuscript has been deemed suitable for publication in PLOS ONE. Congratulations! Your manuscript is now with our production department. 

Kind regards, 

on behalf of

Dr. Ricardo Martinez-Garcia 

Academic Editor

PLOS ONE